environmental chemistry

magnetic sludge biochar, tetracycline, size effect, adsorption

**Authors for correspondence:**
Suxing Luo
e-mail: suxingluo@126.com
Feng Feng
e-mail: feng-feng64@263.net

†These authors contributed equally to this study.

This article has been edited by the Royal Society of Chemistry, including the commissioning, peer review process and editorial aspects up to the point of acceptance.

# Tetracycline adsorption on magnetic sludge biochar: size effect of the Fe₃O₄ nanoparticles

Suxing Luo[1,2,†], Jun Qin[1,†], Yuanhui Wu[2,3] and Feng Feng[1]

[1]College of Chemistry and Environmental Engineering, Shanxi Datong University, Datong 037009, People's Republic of China
[2]Department of Chemistry and Chemical Engineering, Zunyi Normal College, Zunyi 563006, People's Republic of China
[3]Special Key Laboratory of Electrochemistry for Materials of Guizhou Province, Zunyi, 563006, People's Republic of China

SL, 0000-0002-2427-6832

Activated sludge, which is difficult and expensive to treat and dispose of, is a key concern in wastewater treatment plants. In this study, magnetic sludge biochar containing activated sludge and different sizes (14.3, 40.2 and 90.5 nm) of Fe₃O₄ nanoparticles was investigated as an effective adsorbent for tetracycline (TC) adsorption. Magnetic sludge-based biochar was prepared by a facile cross-linking method and characterized by transmission electron microscopy, Fourier transform infrared spectroscopy (FTIR), X-ray diffraction, X-ray photoelectron spectroscopy (XPS) and zeta potential analysis. The adsorption performances of TC on three kinds of adsorbents were investigated. Although 14.3 nm Fe₃O₄ nanoparticles could be inclined to aggregate and partially filled with pores of biochar, it turned out that magnetic sludge biochar with 14.3 nm Fe₃O₄ nanoparticles exhibited optimum performance for TC removal with adsorption capacity up to 184.5 mg g⁻¹, due to the larger amounts of functional groups and the change of zeta potential. Furthermore, the adsorption kinetics of TC on three kinds of adsorbents were studied, which implied that the pseudo-second-order kinetic model exhibited the better fit for the entire sorption process.

## 1. Introduction

The extensive use of antibiotics has become a serious environmental problem, which could induce the destruction of water systems and even in soil [1,2]. Among the commonly

**Scheme 1.** Schematic illustration of the synthesis of MS with different sized Fe₃O₄ particles and its adsorption properties of TC.

used antibiotics, tetracycline (TC) is the most widely used in aquaculture and pharmacy due to the relatively low costs [3,4]. Unfortunately, TC is a typical sterilizing medicine, which is excreted in the form of original drug or parent compounds for humans or animals, and is subsequently discharged into the environment [5,6]. As a result, TC has been detected in wastewater, surface and groundwater, and these trace amounts of TC could cause adverse consequences. For example, TC has been proved to induce micro-bial resistance, resulting in the production of resistant bacteria and resistance genes, and seriously threatening human health [7,8].

Therefore, it is of the utmost importance to eliminate TC from wastewater, and some technologies have been developed, such as adsorption, membrane separation, photocatalysis and chemical oxidation [9–12]. Among these methods, adsorption is still the most promising method due to the cost efficiency, wide range of application, simple design and less toxic byproducts.

Recently, magnetic sludge biochar has attracted much attention in the field of pollutant removal, owing to the high specific surface area, appreciable amounts of active sites and the highly developed pore structure, and could be easily recycled [13,14]. There have been some reports of TC sorption on magnetic sludge biochar. Tang *et al.* reported a novel and effective alkali-acid combined method to prepare magnetic sludge biochar and applied it to TC adsorption. The results showed that the magnetic sludge biochar had high sorption capacities for TC [15]. More recently, Wei *et al.* [16] used the hydrothermal method to synthesis an iron loaded sludge biochar, and then the adsorption behaviour of TC and doxycycline (DOX) onto the as-prepared magnetic sludge biochar was investigated. However, so far, very little attention has been paid to investigations of size-dependent adsorption performance of Fe₃O₄. Possible mechanisms explaining the size effects of Fe₃O₄ on TC adsorption to magnetic sludge biochar could have the effect of promoting or impeding. In a positive way, (i) the increased specific surface area with smaller particle size of Fe₃O₄ could accelerate adsorption ability; (ii) the smaller the particle size of Fe₃O₄, the more surface functional groups would lead to an increase of TC adsorption. In a negative way, (iii) the Fe₃O₄ nanoparticles may result in pores blockage which will lead to a decrease of TC adsorption; (iv) the smaller Fe₃O₄ nanoparticles were inclined to aggregate which could inhibit the adsorption of TC. Moreover, the particle size of Fe₃O₄ may alter the surface properties of adsorbents, such as zeta potential, which could modify the adsorption behaviour of TC. The final outcome, as multiple mechanisms mentioned above acting together, cannot predict straightforwardly.

In the present study, we synthesized magnetic sludge biochar with different sized Fe₃O₄ particles (14.3, 40.2 and 90.5 nm). Furthermore, the adsorption properties of TC were explored by batch sorption experiments, and the process is depicted in scheme 1. This study not only highlights the resource recovery of wastewater treatment sludge for antibiotics removal but also promotes a better understanding of the size effect of Fe₃O₄ particles of magnetic sludge biochar on the sorption of antibiotics.

# 2. Material and methods

## 2.1. Materials

The activated sludge was sampled from Gaoqiao Wastewater Treatment Plant in Zunyi, China. TC (greater than or equal to 98%) and glutaraldehyde (50%) were purchased from Aladdin Industrial

Corporation (USA). $Fe_3O_4$ nanoparticles of different sizes were purchased from Macklin Biochemical Technology Co., Ltd (Shanghai, China). All other chemicals were of analytical grade and were used without any further purification.

## 2.2. Preparation of magnetic sludge biochar (MS)

First, 5 ml 1 mol l$^{-1}$ HCl was added into 500 ml sludge (suspended solid concentration: 14.0 g l$^{-1}$) and stirred for 30 min. To destroy the cell wall, the agitated mixture was transferred to an ultrasonic cell crusher and pretreated under 59 KHz for 15 min, and then dried at 105°C overnight. Second, the dried sludge was calcined at 600°C for 2 h under nitrogen protection with heating rate of 5°C min$^{-1}$. After being cooled to room temperature, the sludge biochar was obtained. Finally, for magnetic sludge biochar with 14.3 nm $Fe_3O_4$ nanoparticles (MS-1), 0.5 g $Fe_3O_4$ nanoparticles (14.3 nm) were added into 100 ml pure water, then 200 ml of 10 g l$^{-1}$ sludge biochar and 80 ml 3% glutaraldehyde were added and the mixture was stirred continuously for 24 h at room temperature [17]. The solid phase was separated by an external magnet and rinsed several times with pure water until the pH was neutral. Afterwards, the magnetic sludge biochar precipitates were dried in a vacuum at 40°C. By the same method, the magnetic sludge biochar with 40.2 and 90.5 nm $Fe_3O_4$ nanoparticles (MS-2, MS-3) were prepared, respectively.

## 2.3. Characterization of magnetic sludge biochar

The morphology of magnetic sludge biochar was determined using a transmission electron microscope TEM (Tecnai G2 F20 S-Twin, FEI, USA). The porosity was determined by $N_2$ adsorption–desorption conducted on BSD-PS (Beishide Instruments, China). X-ray powder diffraction (XRD) patterns of adsorbents were recorded by BrukerAXS (BRUKER, Germany). The surface electronic structure of the adsorbents was analysed by X-ray photoelectron spectroscopy (XPS, Thermo Kalpha, USA). A Fourier transform infrared spectroscopy (FTIR) analyser (VERTEX70 spectrometer, Bruker Co., Germany) was used for FT-IR spectroscopy of the magnetic biochar. The zeta potentials of magnetic biochar were determined using a Malvern Zetasizer (Nano ZS90, Malvern, UK).

## 2.4. Batch sorption experiment

The sorption experiments were conducted in a 150 ml flask by mixing a certain amount of magnetic sludge biochar in TC solution and agitated in the incubator shaker at 150 r.p.m. At certain time intervals, 5.0 ml of the suspension was sampled and separated magnetically. Then, the concentration of TC in the supernatant was determined by UV-Vis (electronic supplementary material, figures S1, S2 and S3).

## 2.5. The regenerability of magnetic sludge biochar

After the sorption experiment was completed, magnetic sludge biochar was separated magnetically. The magnetic sludge biochar was immersed in 0.15 mol l$^{-1}$ NaOH solution, stirred for 3 h at room temperature, then replaced with fresh NaOH solution and stirred for another 3 h for complete desorption.

# 3. Results and discussion

## 3.1. Characterization of magnetic sludge biochar

The morphology of magnetic sludge biochar with different sized $Fe_3O_4$ particles was obtained by TEM, and the particle size was analysed using the Image J software (figure 1). Most of the $Fe_3O_4$ particles distributed on the surface of sludge biochar were in the shape of ellipsoidal in MS-1, MS-2 and MS-3 with the average size of 14.3, 40.2 and 90.5 nm, respectively. Noticeably, the $Fe_3O_4$ particles of MS-1 show the form of aggregation, which suggests smaller $Fe_3O_4$ particles were more likely to aggregate than larger ones.

The surface and pore properties of as-prepared magnetic biochar were characterized by nitrogen adsorption–desorption test. As depicted in figure 2$a$, the isotherm curves of three kinds of adsorbents were consistent with the IUPAC classification type IV curve and $H_3$ type hysteresis loop. The lack of a saturated adsorption platform in the medium relative pressure area indicated that the biochar had

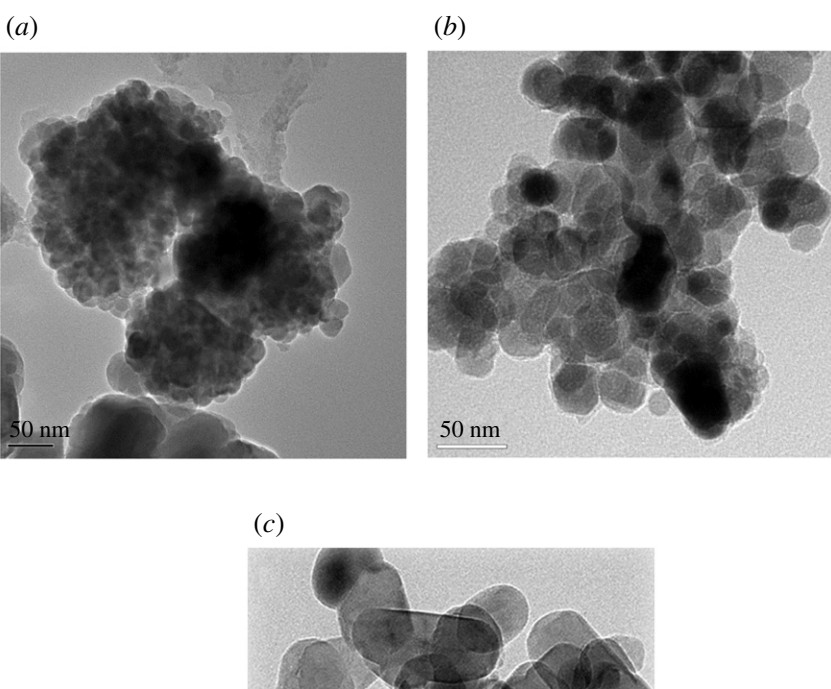

**Figure 1.** TEM analysis of three kinds of magnetic biochar ((a) MS-1; (b) MS-2 and (c) MS-3).

irregular and nonuniform pores and morphologies. The isotherm has no obvious inflection point in the low relative pressure region, indicating that the magnetic biochar has a low proportion of micropore. The specific surface areas of MS-1, MS-2 and MS-3 were calculated as 75.34, 76.27 and 65.45 $m^2 g^{-1}$, and the average pore sizes were 10.43, 11.814 and 11.829 nm, respectively. Compared with MS-2 and MS-3, the specific surface area and the average pore size of MS-1 decreased unexpectedly. This could be attributed to 14.3 nm $Fe_3O_4$ nanoparticles partially blocking the pores of biochar and aggregation.

To reveal the functional groups of the as-prepared three kinds of magnetic biochar, FTIR analysis was performed (figure 2b). Peaks at about 3409 $cm^{-1}$ (–OH stretching), 1048 $cm^{-1}$ (C–O–C) and 543 $cm^{-1}$ (Fe–O vibration) were detected on all three kinds of magnetic biochar [18–23]. The intensity of the peaks for MS-1 was stronger than MS-2 and MS-3 under identical conditions, indicating that the MS-1 possessed larger amounts of functional groups.

Zeta potential is a useful technique for providing important information about the charge carried by the materials. When pH was lower than the isoelectric point ($pH_{IEP}$), materials surface carried positive charge, and the surface displayed negative charge when pH was greater than $pH_{IEP}$. Figure 2c exhibited the surface zeta potentials of three kinds of magnetic biochar. The isoelectric points ($pH_{IEP}$) of MS-1, MS-2 and MS-3 were estimated to be 5.5, 5.1 and 4.9, respectively. Typically, sludge biochar was rich in hydroxyl, carbonyl and carboxyl groups. In addition, $-FeOH_2^+$ groups were dissociated from the surface of $Fe_3O_4$ nanoparticles under lower pH while $FeO^-$ groups emerged at pH above isoelectric point [24]. The absolute value of zeta potential of MS-1 was higher than MS-2 and MS-3, owing to the larger amounts of functional groups on MS-1.

MS-1, MS-2 and MS-3 were characterized by XRD, as shown in figure 2d. The five characteristic diffraction peaks at 30.041, 35.331, 43.11, 53.41, 57.081 and 62.81 correspond to the (220), (311), (400), (511) and (440) lattice planes of $Fe_3O_4$, respectively. Meanwhile, three distinct characteristic peaks of sludge biochar were detected at 20.9, 26.827, 50.370 and 60.254 [17]. These results indicated that $Fe_3O_4$

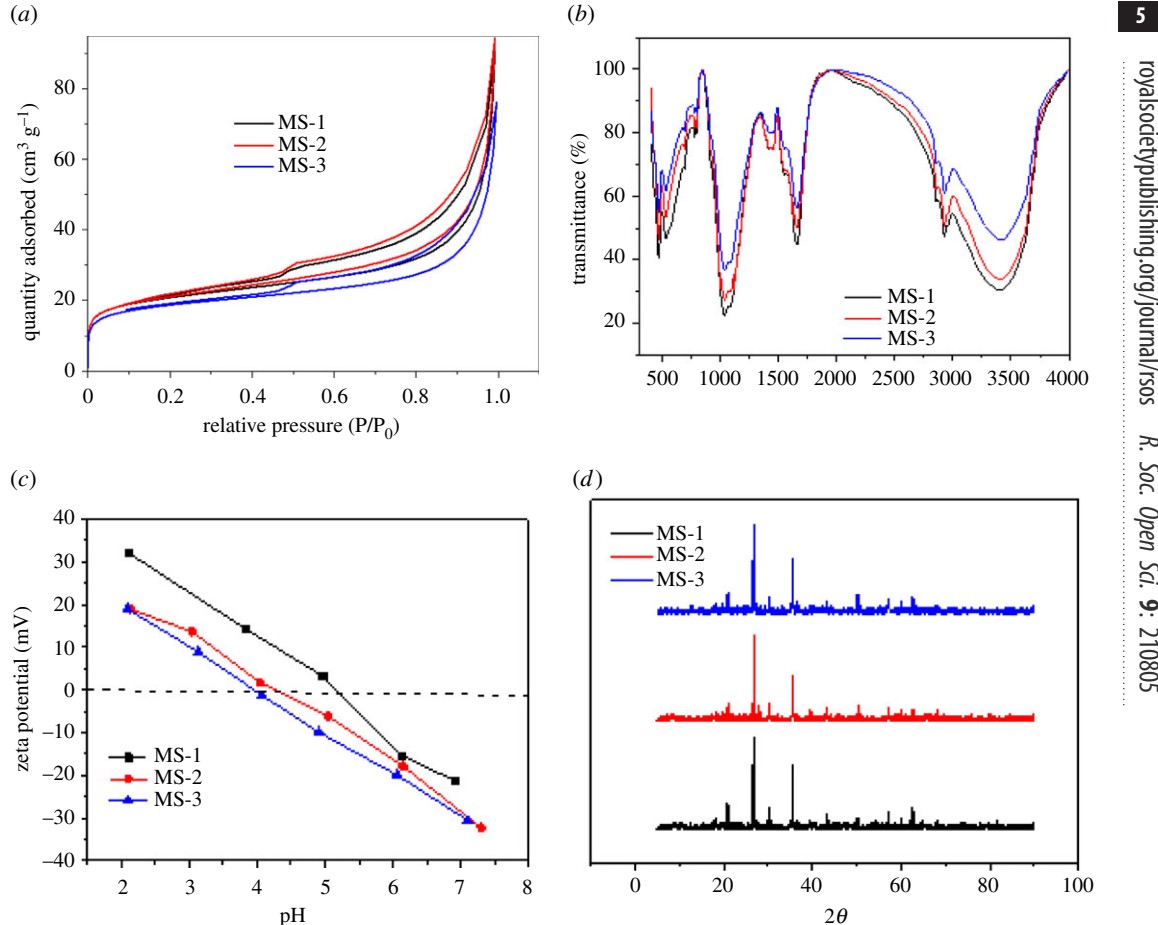

**Figure 2.** Characterizations of the magnetic biochar. (*a*) N$_2$ adsorption–desorption isotherm, (*b*) FT-IR, (*c*) zeta potential and (*d*) XRD patterns.

nanoparticles have been successfully combined with sludge biochar and more importantly, the crystal structures of Fe$_3$O$_4$ and sludge biochar haven't undergone any significant changes.

To confirm the Fe$_3$O$_4$ be synthesized on sludge biochar, XPS was applied to characterize the surface components. Figure 3 presented the XPS spectra of Fe 2p$_{3/2}$ (*a*) and O 1s (*b*) of MS-1, the composition of which was similar to MS-2 and MS-3. The Fe 2p$_{3/2}$ of MS-1 was fitted at 711.3 eV which represents Fe$^{3+}$ in Fe$_3$O$_4$ [25]. Meanwhile, corresponding information revealed in O 1 s spectrum that was fitted to three components at 531.8, 531.2 and 533.2 eV, respectively, the former peak represented the Fe–O–C bond which suggested strong interaction between Fe$_3$O$_4$ and sludge biochar [26]. The later peaks were interpreted as C=O and C–O, respectively, which was attributed to sludge biochar [27].

As observed from a separation test, three kinds of magnetic sludge biochar could be completely separated from TC and MS suspensions within 120 s of application of magnets (electronic supplementary material, figure S4). The results indicated that three kinds of magnetic sludge biochar could be considered as practical adsorbents for removing antibiotics from water and therefore examined subsequently.

## 3.2. Effect of initial pH

Acidity was a crucial parameter that affected the sorption performance of TC, because it could influence the properties of the magnetic sludge biochar and the molecular structural of TC. Hence, the effect of pH on the sorption performance of magnetic sludge biochar was investigated first. Figure 4 presented the sorption capacity at varying pH of the solution from 2.0 to 8.0, and electronic supplementary material, figure S5 showed the existence form of TC at various pH values. The dissociation constants (pKa) of TC are 3.3, 7.7 and 9.7, and the corresponding existing forms in aqueous solution are cation (TC$^+$), molecule (TC$^0$) and anions (TC$^-$ and TC$^{2-}$) [15]. When pH ≤ 3, the sorption capacity of TC on three kinds of adsorbents was very limited, due to both the surface of prepared magnetic biochar and TC

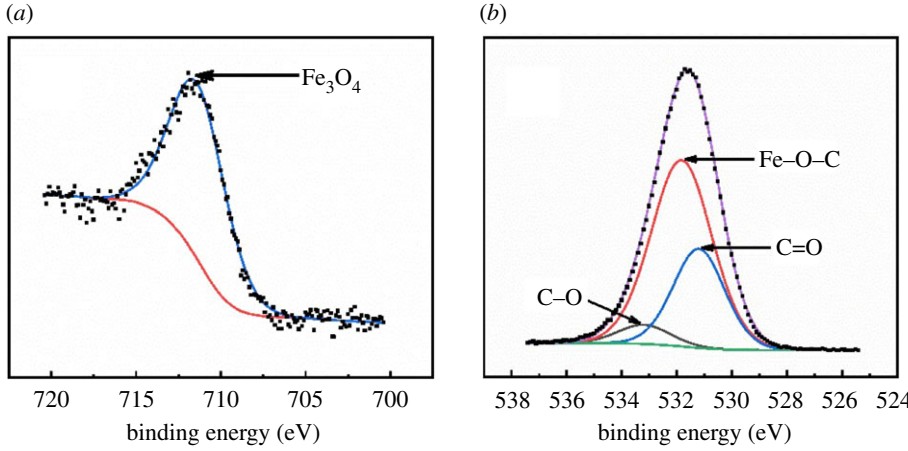

**Figure 3.** XPS spectra of Fe $2p_{3/2}$ (a) and O 1s (b) of MS-1.

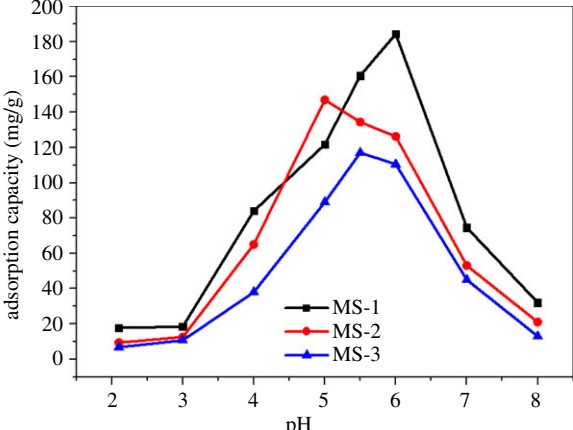

**Figure 4.** Effect of initial pH ($C_0$: 200 mg $l^{-1}$, adsorption time: 1200 min, dosage: 0.65 g $l^{-1}$).

being positively charged. Furthermore, some functional groups of magnetic biochar could be destroyed under the strong acid condition [17]. Along with the increased pH value, the repulsive force slowly decreased, since TC mainly existed as molecules, and the sorption capacity of MS-1, MS-2 and MS-3 increased with pH and reached a maximum at pH = 6.0, 5.0 and 5.5, respectively. When the pH value is close to neutral, although the zeta potential of magnetic biochar was very small and TC mainly existed as neutral molecules, the maximum sorption capacity could be explained by the π–π electron donor–acceptor and hydrogen bonding interactions [16], which was consistent with some previous studies [15,16]. Meanwhile, the adsorption capacities of TC on MS-1, MS-2 and MS-3 were 184.5 mg $g^{-1}$, 134.4 mg $g^{-1}$ and 117.0 mg $g^{-1}$, respectively. This could be attributed to the larger amounts of functional groups of $Fe_3O_4$ particles both on the surface and in the internal pores of sludge biochar than that of MS-2 and MS-3. Nevertheless, when the pH value was greater than 6, the negatively charged magnetic biochar showed an incremental repulsion to anionic $TC^-$ and $TC^{2-}$, along with a theoretical decreasing sorption performance.

## 3.3. Effect of dosage

In the present study, the effect of adsorbent dosage on the sorption of TC was investigated by evaluating the dosage from 0.5 to 0.9 g $l^{-1}$ (figure 5). It was found that the adsorption capacity grew first and then fell with the rise of the dosage. Specifically, the adsorption capacity of TC on MS-2 and MS-3 reached the maximum as the dosage of adsorbent was 0.7 g $l^{-1}$, while the optimum dosage of MS-1 was 0.65 g $l^{-1}$. It was believed that the high MS concentration may result in a cover-up effect on adsorbent surface, blocking the adsorption sites to decrease the adsorption capacity [28,29].

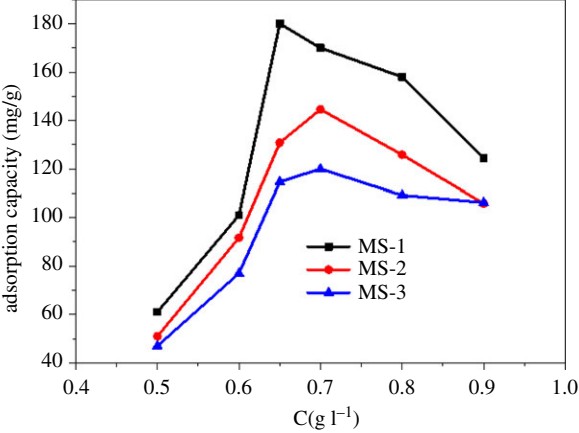

**Figure 5.** Effect of dosage ($C_0$: 200 mg $l^{-1}$, initial pH: 5.5).

**Table 1.** Preparation, and characteristics and adsorption capacity for TC of MS.

| adsorbent | preparation method | magnetic species | functional groups | maximum adsorption capacity for TC | reference |
|---|---|---|---|---|---|
| Fe/Zn magnetic sludge biochar | hydrothermal synthesis | $\gamma$-$Fe_2O_3$ | –OH, C—C, C–O–C, C–H, C—O, Fe–O | 145.0 mg $g^{-1}$ | [30] |
| magnetic cancrinite adsorbent | hydrothermal method | cancrinite | Fe–O | 482.6 mg $g^{-1}$ | [31] |
| $Fe_2O_3$ vested sludge biochar | pyrolysis | $Fe_2O_3$ | –$CH_2$–, C–H, Fe–OH, C–O, C–N, Si–O, C=C, N–H | 286.9 mg $g^{-1}$ | [15] |
| iron loaded sludge biochar | hydrothermal method | FeOOH | O–H, C–H, –$OCH_3$, C–O, C=C, Fe–O | 104.8 mg $g^{-1}$ | [16] |
| magnetic sludge biochar | cross-linking | $Fe_3O_4$ (14.3 nm) | –OH, C=O, Fe–O | 184.5 mg $g^{-1}$ | this work |

The preparation, characteristics, as well as the maximum adsorption capacity of magnetic sludge-based biochar for TC were summarized in table 1.

## 3.4. Effect of contact time and adsorption kinetic

Sorption is a time-dependent process. As shown in figure 6a, the whole adsorption process could be divided into two parts: fast- and slow-adsorption stages, which were separated from the plot at 720 min. Since the sorption kinetic studies could provide useful information on the sorption rate, the pseudo-first-order and pseudo-second-order rate models were employed to fit the experimental data. The pseudo-first-order and pseudo-second-order rate are expressed as equations (3.1) and (3.2):

$$\log\left(q_e - q_t\right) = \log q_e - \frac{k_1}{2.303}t \tag{3.1}$$

and

$$\frac{t}{q_t} = \frac{1}{k_2 q_e^2} + \frac{t}{q_e}, \tag{3.2}$$

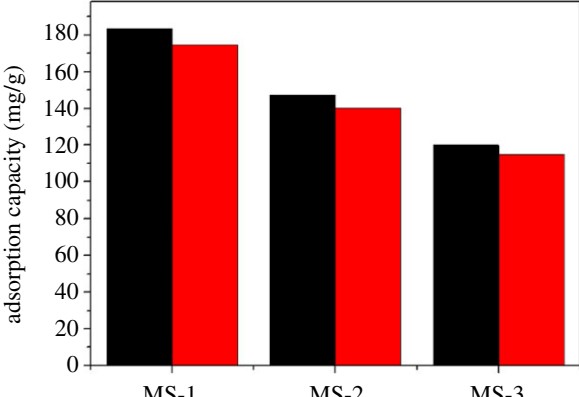

**Figure 6.** (a) Effect of contact time, (b) pseudo-first-order kinetic model and (c) pseudo-second-order kinetic model of TC adsorption on three kinds of magnetic sludge biochar.

**Figure 7.** Regeneration studies of magnetic sludge biochar over four cycles.

where $q_e$ and $q_t$ (mg g$^{-1}$) are the adsorption amounts of TC adsorbed at equilibrium and $t$ (min), respectively. $k_1$ (1 min$^{-1}$) and $k_2$ (g mg$^{-1}$ min) are the rate constants of the pseudo-first order and second order, respectively. $t$ is the contact time (min). Figure 6b,c shows the fitting result of the pseudo-first-order and second-order kinetic models. The R of pseudo-second-order kinetic model was 0.991, 0.994 and 0.996, whereas the R of pseudo-first-order model was 0.985, 0.969 and 0.923 of MS-1, MS-2 and MS-3, respectively. Thus, all the sorption processes could be well expressed by pseudo-second-order kinetic model, demonstrating the sorption processes were much controlled by chemical mechanism [22,23].

## 3.5. The reusability of magnetic sludge biochar

The reusability is one of the most important properties of the adsorbent. After four recycles, the adsorption capacities of MS-1, MS-2 and MS-3 for TC were still as high as 175.5, 130.2 and 114.9, respectively (figure 7). The adsorption capacities decreased slightly in the recycling process which may be due to incomplete desorption of TC which occupied the adsorption sites on MS. Overall, the adsorption capacities of three kinds of adsorbents for TC were still above 95% in the fourth cycle, demonstrating that three kinds of magnetic sludge biochar were reusable as an economical and recyclable magnetic adsorbent for the treatment of antibiotics wastewater.

# 4. Conclusion

In summary, municipal sewage sludge was chosen as the raw material, based on which three kinds of magnetic sludge biochar with different sizes of $Fe_3O_4$ nanoparticles was successfully synthesized and characterized by TEM, FTIR, XRD, XPS and zeta potential analysis. Although 14.3 nm $Fe_3O_4$ nanoparticles could be partially filled into pores of biochar as well as being more likely to aggregate than larger ones, the results demonstrated that magnetic sludge biochar with 14.3 nm $Fe_3O_4$ nanoparticles had larger adsorption capacity for TC than magnetic sludge biochar with 40.2 nm and 90.5 nm $Fe_3O_4$ nanoparticles. These results could be attributed to the fact that $Fe_3O_4$ nanoparticles with a smaller size had larger amounts of functional groups than the larger ones. In addition, experimental parameters such as pH, dosage and contact time were demonstrated to play an important role in the above sorption mechanism and could thus jointly control the sorption behaviours of TC. Besides the effective adsorption capacity for TC, the great regeneration performance and magnetic separation ability entrust magnetic sludge biochar with excellent potential for TC removal. Moreover, the smaller particle size of $Fe_3O_4$ particles leads to the increase of the absolute value of zeta potential and the increase of the isoelectric point, which makes the optimal adsorption conditions closer to the neutral and demonstrates its advantage of applicability in real wastewater treatment. Apparently, size dependency provides a clue for materials modification and supplies a cost-effective way for municipal sewage sludge resource disposal.

Data accessibility. The datasets supporting this article are available from Zenodo: https://zenodo.org/record/5839082#.Yd4wB8gbHJ8. The data are provided in the electronic supplementary material [32].

Authors' contributions. S.L. contributed to the conception of the study and experiment; J.Q. contributed significantly to analysis and manuscript preparation; Y.W. performed the experiment and participated in data analysis; F.F. conceived of the study, designed the study, coordinated the study and helped draft the manuscript. All authors gave final approval for publication and agreed to be held accountable for the work performed therein.

Competing interests. We declare we have no competing interests.

Funding. This work was supported by the Open Fund of Innovation and Application Engineering Research Center for Mesoporous Materials of Shanxi Province (no. MMIA2019002), the Research Fund for the Doctoral Program of Zunyi Normal College (no. BS [2019]13), the Special Key laboratory of Electrochemistry for Materials of Guizhou Province (no. KY [2018]004), the Zunyi City-School Joint Fund (no. HZ [2021]198), the Applied Basic Research Project of Shanxi Province (no. 201801D221056), the Applied Basic Research Program of Datong (no. 2018147) and the Scientific and Technological Innovation Programs of Higher Education Institutions in Shanxi (2020L0498).

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
