## [Peer Review File · Royal Society Open Science]

Review History

RSOS-210805.R0 (Original submission)

Review form: Reviewer 1

Is the manuscript scientifically sound in its present form?

No

Are the interpretations and conclusions justified by the results?

Yes

Is the language acceptable?

Yes

Do you have any ethical concerns with this paper?

No

Have you any concerns about statistical analyses in this paper?

No

Recommendation?

Major revision is needed (please make suggestions in comments)

Comments to the Author(s)

I have read the manuscript entitled "Tetracycline Adsorption on Magnetic Sludge Biochar: size effect of the Fe₃O₄ nanoparticles" by Luo et al. to Royal Society Open Science.

The aim of this manuscript was to prepare and characterize magnetic sludge biochar with 15 nm, 50 nm and 100 nm Fe₃O₄ nanoparticles and to analyze the adsorption performance of tetracycline on the these 3 kinds of adsorbents.

In my opinion, it is not clear in the text what is the real contribution of this manuscript compared to other works in the literature. The size-dependent adsorption performance of Fe₃O₄ in my opinion is not a strong contribution. It is intrinsic that the reduction in particle size will increase the adsorption capacity.

Page 8 - Figure 4 - The authors should further explain why the adsorption capacity decreases with increasing TC dosage.

See this paper:

Lima, M.M., Macuvele, D.L.P., Nones, J. et al. Synthesis of Fe₃O₄-Fe₂O₃@C Core-Shell Nanoparticles: Effect of Reactional Parameters on Structural and Magnetics Properties. *J Inorg Organomet Polym* 29, 1848-1861 (2019). <https://doi.org/10.1007/s10904-019-01146-8>

Review form: Reviewer 2 (Bhaskar Sathe)

Is the manuscript scientifically sound in its present form?

No

Are the interpretations and conclusions justified by the results?

No

Is the language acceptable?

No

Do you have any ethical concerns with this paper?

No

Have you any concerns about statistical analyses in this paper?

Yes

Recommendation?

Major revision is needed (please make suggestions in comments)

Comments to the Author(s)

1. Author need to work on introduction to reflect the novelty of the proposed system and urgency of the studies.
2. The proposed methodology is not clear, how selectively 15, 50, 100 nm Fe₃O₄ is formed?
3. TEM need to do again with low magnification to show homogeneous and know more yield of nanoparticles.

4. XRD, XPS is required to know impurities and phases for all three samples.
5. Why pH -7 is better adsorption?
6. Compare the performance of adsorption with similar systems from literature (comparative table).
7. Author need to give clarification on major decrement in performance with cycle number drastically.

Decision letter (RSOS-210805.R0)

Dear Dr Luo:

Title: Tetracycline adsorption on magnetic sludge biochar: size effect of the Fe₃O₄ nanoparticles
Manuscript ID: RSOS-210805

The editor assigned to your manuscript has now received comments from reviewers. We would like you to revise your paper in accordance with the referee and Subject Editor suggestions which can be found below (not including confidential reports to the Editor). Please note this decision does not guarantee eventual acceptance.

Please submit your revised paper before 22-Oct-2021. Please note that the revision deadline will expire at 00.00am on this date. If we do not hear from you within this time then it will be assumed that the paper has been withdrawn. In exceptional circumstances, extensions may be possible if agreed with the Editorial Office in advance. We do not allow multiple rounds of revision so we urge you to make every effort to fully address all of the comments at this stage. If deemed necessary by the Editors, your manuscript will be sent back to one or more of the original reviewers for assessment. If the original reviewers are not available we may invite new reviewers.

Yours sincerely,
Dr Ellis Wilde
Publishing Editor, Journals

On behalf of the Subject Editor Professor Anthony Stace and the Associate Editor Dr Dattatray Late.

RSC Associate Editor
Comments to the Author:
Major Revision needed.

RSC Subject Editor
Comments to the Author:
(There are no comments.)

Reviewers' Comments to Author:

Reviewer: 1

Comments to the Author(s)

I have read the manuscript entitled "Tetracycline Adsorption on Magnetic Sludge Biochar: size effect of the Fe₃O₄ nanoparticles" by Luo et al. to Royal Society Open Science.

The aim of this manuscript was to prepare and characterize magnetic sludge biochar with 15 nm, 50 nm and 100 nm Fe₃O₄ nanoparticles and to analyze the adsorption performance of tetracycline on the these 3 kinds of adsorbents.

In my opinion, it is not clear in the text what is the real contribution of this manuscript compared to other works in the literature. The size-dependent adsorption performance of Fe₃O₄ in my opinion is not a strong contribution. It is intrinsic that the reduction in particle size will increase the adsorption capacity.

Page 8 - Figure 4 - The authors should further explain why the adsorption capacity decreases with increasing TC dosage.

See this paper:

Lima, M.M., Macuvele, D.L.P., Nones, J. et al. Synthesis of Fe₃O₄-Fe₂O₃@C Core-Shell Nanoparticles: Effect of Reactional Parameters on Structural and Magnetics Properties. J Inorg Organomet Polym 29, 1848-1861 (2019). <https://doi.org/10.1007/s10904-019-01146-8>

Reviewer: 2

Comments to the Author(s)

1. Author need to work on introduction to reflect the novelty of the proposed system and urgency of the studies.
2. The proposed methodology is not clear, how selectively 15, 50, 100 nm Fe₃O₄ is formed?

3. TEM need to do again with low magnification to show homogeneous and know more yield of nanoparticles.
4. XRD, XPS is required to know impurities and phases for all three samples.
5. Why pH -7 is better adsorption?
6. Compare the performance of adsorption with similar systems from literature (comparative table).
7. Author need to give clarification on major decrement in performance with cycle number drastically.

Author's Response to Decision Letter for (RSOS-210805.R0)

See Appendices A & B.

RSOS-210805.R1 (Revision)

Review form: Reviewer 1

Is the manuscript scientifically sound in its present form?

Yes

Are the interpretations and conclusions justified by the results?

Yes

Is the language acceptable?

Yes

Do you have any ethical concerns with this paper?

No

Have you any concerns about statistical analyses in this paper?

No

Recommendation?

Accept as is

Comments to the Author(s)

Dear Authors,

I read carefully again the paper entitled "Tetracycline Adsorption on Magnetic Sludge Biochar: size effect of the Fe₃O₄ nanoparticles" by Luo et al. to Royal Society Open Science.

Based on this I am satisfied with the authors response to the questions that I raised in the first revision round.

Review form: Reviewer 2

Is the manuscript scientifically sound in its present form?

Yes

Are the interpretations and conclusions justified by the results?

Yes

Is the language acceptable?

Yes

Do you have any ethical concerns with this paper?

No

Have you any concerns about statistical analyses in this paper?

No

Recommendation?

Accept as is

Comments to the Author(s)

Work is now acceptable for publication

Decision letter (RSOS-210805.R1)

Dear Dr Luo:

Title: Tetracycline adsorption on magnetic sludge biochar: size effect of the Fe₃O₄ nanoparticles
Manuscript ID: RSOS-210805.R1

It is a pleasure to accept your manuscript in its current form for publication in Royal Society Open Science. The chemistry content of Royal Society Open Science is published in collaboration with the Royal Society of Chemistry.

Yours sincerely,
Dr Ellis Wilde
Publishing Editor, Journals

On behalf of the Subject Editor Professor Anthony Stace and the Associate Editor Dr Dattatray
Late.

RSC Associate Editor
Comments to the Author:
Authors have revised the manuscript as per comments.

Reviewer(s)' Comments to Author:

Appendix A

Response to Reviewer 1 Comments

We are truly grateful for the reviewer's thoughtful suggestions, and would like to present our point-by-point responses to the reviewers' comments below.

1. In my opinion, it is not clear in the text what is the real contribution of this manuscript compared to other works in the literature. The size-dependent adsorption performance of Fe_3O_4 in my opinion is not a strong contribution. It is intrinsic that the reduction in particle size will increase the adsorption capacity.

The novelty and urgency of the studies have been illustrated in Introduction. Possible mechanisms explaining the size effects of Fe_3O_4 on TC adsorption to magnetic sludge biochar including specific surface area, surface functional groups, pores blockage, aggregation, the surface properties would jointly control the sorption behaviors of TC. Therefore, the final outcome, as multiple mechanisms mentioned above acting together, can not predict straightforwardly.

In this paper, we discussed the adsorption behavior of TC on magnetic sludge biochar with different particle size of Fe_3O_4 under the influences of various factors, and the particle size of Fe_3O_4 nanoparticles was optimized. Furthermore, the results provide a clue for materials modification, and supplies a cost-effective way for municipal sewage sludge resource disposal.

2. Page 8 - Figure 4 - The authors should further explain why the adsorption capacity decreases with increasing TC dosage.

This is a very helpful suggestion. We have explained the reason for adsorption capacity decreases with increasing TC dosage in Section 3.3. "It was believed that the high MS concentration may result in a cover-up effect on adsorbent surface, blocking the adsorption sites to decrease the adsorption capacity".

Appendix B

Response to Reviewer 2 Comments

We greatly thank the reviewer for the helpful comments and constructive suggestions, and would like to present our point-by-point responses to the reviewers' comments below.

1. Author need to work on introduction to reflect the novelty of the proposed system and urgency of the studies.

The reviewer's suggestion is quite valuable. The novelty of the proposed system and urgency of the studies have been discussed in Introduction in page 2.

2. The proposed methodology is not clear, how selectively 15, 50, 100 nm Fe₃O₄ is formed?

Different size of Fe₃O₄ particles were purchased from Macklin Biochemical Technology Co., Ltd (Shanghai, China) which mentioned in Section 2.1. The actual size of Fe₃O₄ particles was obtained by TEM and the particle size was analyzed using the Image J software (Fig. 1). The average sizes of Fe₃O₄ on MS-1, MS-2, MS-3 were 14.3, 40.2, 90.5 nm, respectively.

We apologize for our negligence and have added the actual size of Fe₃O₄ particles accordingly in our manuscript.

3. TEM need to do again with low magnification to show homogeneous and know more yield of nanoparenticles.

We agree with reviewer and the TEM has been done again and discussed in detailed in Section 3.1.

4. XRD, XPS is required to know impurities and phases for all three samples.

This is again a very helpful suggestion. XRD and XPS were applied to characterize the magnetic sludge-based biochar, and the results were shown in Fig. 2(D) and Fig. 3, respectively. The phases were discussed in detail in Section 3.1.

5. Why pH -7 is better adsorption?

The sorption capacity of MS-1, MS-2, MS-3 increased with pH and reached a maximum at pH=6.0, 5.0, 5.5, respectively. The maximum sorption capacity could be explained by the π - π electron donor-acceptor and hydrogen bonding interactions, which was consistent with some previous studies. The results were discussed in detailed in Section 3.2.

6. Compare the performance of adsorption with similar systems from literature

(comparative table).

The preparation, characteristics, as well as the maximum adsorption capacity of magnetic sludge-based biochar for TC were summarized in Table 1 (section 3.3).

7. Author need to give clarification on major decrement in performance with cycle number drastically.

After four recycles, the adsorption capacities of MS-1, MS-2 and MS-3 for TC were still as high as 175.5, 130.2, and 114.9 mg/g, respectively (Fig.7). Overall, the adsorption capacities of three kinds of adsorbents for TC were still above 95% in the fourth cycle. The adsorption capacities decreased slightly in the recycling process may due to incomplete desorption of TC which occupied the adsorption sites on MS. This is available at Section 3.5.